# Transconjunctival versus Transcutaneous Injection of Botulinum Toxin into the Lacrimal Gland to Reduce Lacrimal Production: A Randomized Controlled Trial

**DOI:** 10.3390/toxins13020077

**Published:** 2021-01-21

**Authors:** Andrew G. Lee, Shin-Hyo Lee, Minsu Jang, Sang Jae Lee, Hyun Jin Shin

**Affiliations:** 1Department of Ophthalmology, Blanton Eye Institute, Houston Methodist Hospital, Houston, TX 77030, USA; AGLee@houstonmethodist.org; 2Department of Ophthalmology, Neurology, Neurosurgery, Weill Cornell Medicine, New York, NY 10021, USA; 3Department of Ophthalmology, University of Texas Medical Branch, Galveston, TX 77555, USA; 4Department of Ophthalmology, UT MD Anderson Cancer Center, Houston, TX 77030, USA; 5Department of Ophthalmology, Texas A and M College of Medicine, College Station, TX 77807, USA; 6Department of Ophthalmology, University of Iowa Hospitals and Clinics, Iowa City, IA 52242, USA; 7Department of Ophthalmology, Baylor College of Medicine and the Center for Space Medicine, Houston, TX 77030, USA; 8Department of Ophthalmology, University of Buffalo, Buffalo, New York, NY 14214, USA; 9Department of Anatomy, Yonsei University College of Medicine, Seoul 03722, Korea; veterinarian2@hanmail.net; 10Department of Ophthalmology, Research Institute of Medical Science, Konkuk University Medical Center, Konkuk University School of Medicine, Seoul 05030, Korea; jmsmed12@naver.com; 11School of Medicine, Konkuk University, Seoul 05029, Korea; atpsang@naver.com

**Keywords:** botulinum toxin, delivery route, epiphora, lacrimal gland, lacrimal obstruction

## Abstract

The purpose of this study was to determine and compare the effects between injecting botulinum toxin A (BTX-A) transconjunctivally into the palpebral lobe and transcutaneously into the orbital lobe of the lacrimal gland in patients with epiphora due to lacrimal outflow obstruction. This randomized controlled study included 53 eyes of 31 patients with unilateral or bilateral epiphora. Patients were randomly allocated to receive an injection of BTX-A (3 units) either transconjunctivally (*n* = 15, 25 eyes) or transcutaneously (*n* = 16, 28 eyes). For objective assessments, the tear meniscus height and Schirmer’s I test with topical anesthesia were measured at baseline and after 2, 6, 12, and 24 weeks of follow-up. Subjective evaluations were performed using the Munk score. After BTX-A injection, patients in both groups experienced significant objective and subjective reductions in tearing at all follow-up times compared to pre-injection (success rate 86.8%), and the effect lasted for a mean duration of 5.63 months. The two delivery routes showed similar clinical effectiveness for a single injected dose of BTX-A. In conclusion, injecting BTX-A via either a transconjunctival or transcutaneous route helps to reduce normal tear production and results in significant improvements in the symptoms in patients with epiphora.

## 1. Introduction

Epiphora is a frequent complaint in ophthalmology clinics, and can affect the quality of life by causing blurry vision, periocular irritation, and social embarrassment. Lacrimal outflow obstruction is one of the most common causes of epiphora, which is usually treated using intubation by a silicone tube for the functional obstruction or dacryocystorhinostomy for obstruction of the lower system or a Jones tube for obstruction of the upper system [1]. However, epiphora can reoccur immediately after removing the silicone tube [2], while dacryocystorhinostomy failures reportedly occur in 10–20% of cases. Some patients complain of persistent epiphora even when anatomical success occurs [3], while inserting a Jones tube is associated with a high rate of tube displacement [4].

Botulinum toxin A (BTX-A) is a neurotoxin that reversibly blocks the release of acetylcholine from cholinergic nerve terminals of neuromuscular synapses and autonomic nerve fibers. This neurotoxin has been used to weaken overactive muscles and to control hypersecretion from glands supplied by cholinergic neurons [5,6]. Injecting BTX-A directly into the lacrimal gland has also been shown to be effective for reducing the hyperlacrimation associated with gustatory tearing [7]. Since injecting BTX-A is technically easy, minimally invasive, and induces reversible effects, it can be considered as an alternative treatment in patients with lacrimal outflow obstruction, especially for surgery candidates in a poor systemic condition such as due to being elderly and having comorbidities [8,9]. 

BTX-A has been injected both transconjunctivally into the palpebral lobe of the lacrimal gland and transcutaneously into the orbital lobe [10,11]. Both delivery routes provide symptom relief with minimal complications. However, the effectiveness and side effects have not been systematically compared between these two injection routes. This randomized clinical trial therefore aimed to establish the efficacy and safety of BTX-A injections into the lacrimal gland and compared the two delivery routes in patients with epiphora due to anatomical or functional lacrimal outflow obstructions.

## 2. Results

The findings of objective and subjective evaluations of the transconjunctival (CON) and transcutaneous (CUT) injection groups after injecting BTX-A are compared in Figure 1, Figure 2 and Figure 3 and Appendix A.

### 2.1. Objective Outcome

The tear meniscus height (TMH) and Schirmer’s I test with topical anesthesia (STA) value decreased significantly in both groups compared to baseline during the 24 weeks of follow-up (all *p* < 0.05) (Figure 1 and Figure 2). The TMH was lower in the CON group than the CUT group throughout the first 12 weeks of follow-up. Repeated-measures analysis of variance (RMANOVA) demonstrated no significant differences between the two groups in either the TMH or STA value over time (*p* = 0.218 and *p* = 0.459, respectively). Schirmer’s I test without topical anesthesia measuring both basal and reflex tears secretion was also reduced after BTX-A injection (Appendix A).

### 2.2. Subjective Outcome

Both groups exhibited significant improvements in outdoor and indoor Munk scores compared to baseline during the 24 weeks of follow-up (all *p* < 0.05), while there were no significant differences between the two groups over time (*p* = 0.218 and *p* = 0.082, respectively) (Figure 3). The mean Glasgow Benefit Inventory (GBI) score during the follow-up was 16.15 (95% confidence interval [CI] = 11.06–1.24) in the CON group and 13.09 (95% CI = 8.48–17.69) in the CUT group, which demonstrated positive changes in the health-related quality of life for both intervention groups. The biggest improvements were in the overall quality of life (3.65), feeling more optimistic about the future (3.65), and feeling better about yourself (3.63) (Appendix A showing the GBI subscale). The GBI score did not differ significantly between the two groups over time (*p* = 0.415). The duration of symptom relief in both groups was 5.63 ± 2.05 months, ranging from 1 to 10 months, and did not differ significantly between the two groups (5.25 ± 2.42 months in the CON group and 6.07 ± 1.48 months in the CUT group, *p* = 0.176).

### 2.3. Overal Success and Adverse Effect

The overall success rate in the CON group was 89.3% (*n* = 13, 25 eyes). Two patients (7.1%, two eyes) had blepharoptosis that resolved at 3–4 weeks after the injection. One patient (3.6%, one eye) complained of blepharoptosis and mild binocular diplopia, both of which resolved within 4 weeks. Eleven patients (75%, 21 eyes) responded that they recommended the treatment to others. The overall success rate in the CUT group was 84% (*n* = 13, 21 eyes). One patient (8%, two eyes) had blepharoptosis that resolved within 4 weeks after the injection. One patient (4%, one eye) complained of mild dry-eye symptoms that completely resolved after using artificial eye drops. Eleven patients (72%, 18 eyes) responded that they recommended the treatment to others. There were no adverse events related to vital signs or in physical and neurologic examinations and laboratory tests. The adverse effect, recommendation rate, and overall success did not differ significantly between the two groups (*p* = 1.000, *p* = 1.000, and *p* = 0.694, respectively).

## 3. Discussion

In our cohort of patients, the overall success rate among the enrolled patients was 86.8% with a good safety profile, and the effect of each injection lasted for a mean of 5.63 months. Furthermore, patient satisfaction seemed to be high, as indicated by the relatively high rate of recommendations to other patients (73.6%). The two delivery routes showed similar clinical effectiveness for a single injected dose of BTX-A. The results obtained in this study are similar to those of previous studies of BTX-A injections for epiphora in patients with lacrimal outflow obstructions. Girard et al. [12] reported that 90% of patients among 27 cases of obstructive epiphora were very satisfied, with few side effects. Ziahosseini et al. [13] found that 20 of 22 cases (89%) of an anatomical obstruction responded to a mean of 3.5 injections of BTX-A. Also, Whittaker et al. [9] reported epiphora reduction in 12 of 14 eyes (86%) with functional epiphora lasting for up to 13 weeks, after receiving a median dose of 2.5 units. We observed successful outcomes in 9 of the 10 cases (90%) with an anatomical obstruction and 37 of the 43 cases (86.1%) with a functional obstruction. BTX-A injections into the lacrimal gland have previously mainly been used to treat gustatory hyperlacrimation resulting from aberrant regeneration after facial nerve injuries. Thus, the results obtained in this study provide more evidence for the value of BTX-A in the nonsurgical treatment of lacrimal outflow obstruction with normal tear production.

With regard to the injection route, we found no evidence for previous claims that transconjunctival injections are more effective and safer than transcutaneous injections. It has been suggested that transconjunctival injections are preferable due to the ability to directly visualize the lacrimal gland when performing the injection [7,10]. In our randomized control trial, the main measurement outcomes (TMH, STA value, and Munk and GBI scores) showed tendencies toward better results in the CON group than the CUT group, but these differences did not achieve statistical significance over time. Regarding procedural convenience, a transcutaneous injection is easier and more tolerable to the patient, whereas a transconjunctival injection is associated with patient discomfort during lid eversion with the retractor and when applying pressure at the lateral canthal area and eyeball to expose the lacrimal gland.

Regarding the adverse effects of BTX-A injections, our results are consistent with previous reports of about 10% of patients experiencing temporary ptosis and diplopia that typically last for 2–4 weeks [8,9,14]. Also, there have been reports that performing a transcutaneous injection without direct visualization of the lacrimal gland might increase the risk of paresis of the levator palpebrae superioris and extraocular muscles [15,16]. However, in our study, temporary ptosis and diplopia occurred in 10.7% and 8% of cases in the CON and CUT groups, respectively, with no significant intergroup difference. These discrepancies might be attributable to differences in injection techniques and doses. In our protocol we aim the needle directly posteriorly for the first 2 mm and then direct it superotemporally. The needle is advanced 10 mm through the subcutaneous tissue. After the needle tip touches the lacrimal gland fossa, we inject a minimal dose of BTX-A (0.1 mL, 3 units) as we withdraw the needle by about 1 mm (so that the needle is no longer touching the bone) (Figure 5B) [11].

Despite the significant decrease in STA values, there were no side effects in either group in the form of corneal staining or punctuate epitheliopathy with fluorescein staining during this study. After injecting BTX-A, the TMH of the enrolled patients decreased significantly at the follow-up visits (0.20–0.24 mm) to remain around the normal reported value of 0.22 mm [17]. The TMH did not decrease to markedly below the normal value in any of the investigated eyes, which might explain why dry-eye disease was rarely observed. This finding is consistent with Nava-Castaneda et al. [18] reporting that applying a dose of 2.5 units of BTX-A did not cause any complication associated with a decrease in the tear volume.

It is important to consider the injection dose since there is a wide range of different doses for BTX-A injections reported in the published literature (2.5–60 units). However, higher doses seem to provide no additional benefits in terms of efficacy and duration [15,19]. The minimum dose reported to have successfully reduced tears in functional outflow obstruction was 2.5 units [9]. Ziahosseini et al. [13] similarly recommended a starting dose of 2.5 units of BTX-A in patients with canalicular obstruction. In our patient with both anatomical and functional outflow obstructions, injecting 3 units of BTX-A achieved a successful outcome with minimal complications via either a transconjunctival or transcutaneous route.

We found that injecting BTX-A into the lacrimal gland seemed to produce a longer-lasting effect (5.6 months) than the recovery of orbicularis function after injecting BTX-A for blepharospasm. The effect of BTX-A usually subsides after 2–4 months because of the collateral sprouting of nerve endings in muscles, but this may take much longer in the autonomic nervous system [20]. Also, there is less motion in the orbit where the lacrimal gland is located than in the soft tissues of the face, where continuous dynamic motion occurs, which might reduce the rate of BTX-A clearance. These results are supported by previous reports of the transcutaneous injection of 3.3 units of BTX-A providing significant relief from epiphora for 6–8 months [11]. Also, Nava-Castaneda et al. [18] reported that the effects of BTX-A lasted for up to 6 months when it was injected into the lacrimal gland for treating hyperlacrimation as gustatory tearing. These results suggest that patients with lacrimal outflow obstruction who are only troubled significantly during colder weather may only require very infrequent injections and could be good candidates for BTX-A treatment.

Notwithstanding the above advantages of BTX-A, there are concerns about its long-term efficacy and safety. BTX-A has been found to be both safe and effective at the neuromuscular junction, but its long-term effects on the peripheral autonomic system are unknown. It is plausible that the minor trauma associated with repeated injections could eventually impair the function of the lacrimal gland [21]. However, structural changes in the lacrimal gland have not been demonstrated in animal studies [20,22]. Also, Barañano and Miller [21] reported that repeated injections of BTX-A continued to be effective in controlling crocodile tears syndrome over three years without complications. Although no long-term adverse effects have been identified, in addition to the benefits, the cost of repeated injections and potential transient side effects such as diplopia and ptosis should be discussed with the patient before starting treatment.

This study was subject to several limitations, including enrolling a relatively small number of patients and most of them being elderly individuals. Considering that Eustis and Babiuch treated three children with canalicular obstruction over a 2-year period with repeated BTX-A injections without any complications [23], the present findings should also be applicable to young patients. In addition, the effects of BTX-A continued beyond the 24-week study period in some patients, and so a sufficiently long follow-up is necessary to determine the maximum duration of the effect in future studies with larger case series. Long follow-up is necessary to determine the maximum duration of the effect in future studies with preferably a larger number of patients.

## 4. Conclusions

In conclusion, injecting BTX-A into the lacrimal gland relieved the signs and symptoms of epiphora and also improved the quality of life in patients with an anatomical or functional lacrimal outflow obstruction. Either delivery route—transconjunctival injection into the palpebral lobe of the lacrimal gland or transcutaneous injection into the orbital lobe—can effectively relieve epiphora with minimal complications.

## 5. Materials and Methods

This prospective, randomized single-blinded study was conducted at the Department of Ophthalmology, Konkuk University Medical Center, Seoul, Korea between May 2017 and December 2019. This clinical study was approved by the Institutional Review Board/Ethics Committee of Konkuk University Medical Center (registration number: KUH1100045, date of approval 24 November 2016). The study was conducted according to the principles expressed in the Declaration of Helsinki, with informed written consents obtained from all of the participants.

The study inclusion criteria were (1) presence of epiphora caused by an anatomical or functional outflow obstruction and the patient not desiring surgical therapy, (2) persistent epiphora lasting >3 months, (3) no symptom improvement after more than 1 month of nonsurgical treatment, (4) outdoor Munk score of ≥2, and (5) aged >19 years. The exclusion criteria were (1) neuromuscular disorders (e.g., Parkinson’s disease or myasthenia gravis), (2) taking medications known to interfere with the effects of BTX-A within the previous 1 month (e.g., aminoglycoside or benzodiazepines), (3) BTX-A injection within the previous 3 months, (4) previous history of hypersensitivity reactions to BTX-A, (5) eyelid abnormalities (e.g., ptosis or entropion), (6) dysfunction of tear production or secretion (e.g., meibomian gland dysfunction or Sjogren’s syndrome), (7) reflex tearing secondary to dry eye, or (8) any major oculomotor imbalance or ocular pathology.

All of the included patients underwent a comprehensive ophthalmic examination that included a slit-lamp examination, diagnostic nasolacrimal irrigation and probing, the fluorescein dye disappearance test, and an intranasal examination. Functional outflow obstruction was defined as patent irrigation and delayed fluorescein dye disappearance test (FDDT) without any ocular surface disease/structural eyelid abnormality that could be attributed to causing epiphora. Also, anatomical outflow obstruction was defined as patients with punctal stenosis, canalicular obstruction, and nasolacrimal duct obstruction.

The included subjects were randomly divided into the CON and CUT injection groups in a 1:1 ratio. Random numbers were generated by a computerized number generator using SAS software (version 9.3, SAS Institute, Cary, NC, USA). Of 32 patients enrolled, 31 patients (53 eyes) completed the 24-week visit, comprising 16 patients with 28 eyes in the CON group and 15 patients with 25 eyes in the CUT group (Figure 4). They were aged 71.5 ± 7.9 years, ranging from 50 to 90 years. The baseline characteristics of age, sex, obstruction type, laterality, TMH, STA value, and outdoor and indoor Munk scores did not differ significantly between the groups. The patient demographic characteristics and baseline characteristics are summarized in Table 1.

### 5.1. BTX-A Injection into the Lacrimal Gland

Fifty units of BTX-A (Prabotulinumtoxin A; Daewoong Pharmaceuticals Ltd., Seoul, Korea) was reconstituted in 1.6 mL of 0.9% sodium chloride solution. A single dose of 3 units (0.1 mL) of BTX-A was delivered transconjunctivally into the palpebral lobe or transcutaneously into the orbital lobe of the lacrimal gland using a 30-gage, 1/2-inch needle attached to a 1-mL tuberculin syringe (Figure 5) [10,11]. A BTX-A injection was performed either unilaterally or bilaterally according to the epiphora symptoms in each patient (3 units per eye). All BTX-A injections were performed by a single clinician (H.J.S.).

### 5.2. Objective Evaluations

Objective evaluations were performed by (1) measuring the TMH and then (2) performing STA. The tear meniscus was photographed using a digital camera attached to a slit-lamp microscope at the primary position, and the TMH was measured in pixels and converted into millimeters according to a previously reported method [17]. The STA was performed with the instillation of a topical anesthetic (0.5% proparacaine hydrochloride; Alcaine, Alcon, Fort Worth, TX, USA). Briefly, a Schirmer strip was placed on the lower conjunctival sac between the lateral area and the external third of the sac for 5 min to measure the amount of tears produced in millimeters.

### 5.3. Subjective Evaluations

Subjective evaluations were carried using (1) Munk scores (indoor and outdoor) and (2) a quality-of-life questionnaire (Glasgow Benefit Inventory [GBI]). The Munk score used to assess the severity of epiphora in the case group was assigned as follows: 0, no epiphora; 1, epiphora requiring dabbing less than twice daily; 2, epiphora requiring dabbing 2–4 times/day; 3, epiphora requiring dabbing 5–10 times/day; 4, epiphora requiring dabbing >10 times/day; and 5, constant tearing. The GBI is a postinterventional questionnaire assessing the general perception of well-being and changes in social and physical health, and consists of 18 questions with responses scored on a 5-point Likert scale (Appendix A) [24]. Raw response scores on the GBI are converted to a Rasch scale from −100 (maximum detriment) through 0 (no benefit/change) to +100 (maximum benefit). Patient satisfaction was assessed at the last visit by asking them the following question: ‘Do you recommend this treatment to others?’

### 5.4. Follow-Up after Intervention

All of the enrolled patients received a comprehensive ophthalmologic examination and laboratory screening tests before and after the BTX-A injections. Complete subjective and objective evaluations and registration of the adverse events were performed at 2, 6, 12, and 24 weeks after the intervention, with the exception the GBI which was skipped at the 2-week visit. If a BTX-A injection into the lacrimal gland still significantly improved epiphora at the 24-week follow-up visit, then the patient was asked to visit the clinic 3 months later or report how long the reduction of their epiphora symptoms continued. We defined that overall success occurred when the patient was satisfied with the treatment, along with an improvement in both objective and subjective measurements over 6 weeks. All measurements were performed by another examiner (M.J.) who was blinded to the information of the patients.

### 5.5. Statistical Analysis

Data were analyzed using SPSS for Windows (version 24.0, SPSS, Chicago, IL, USA). The Shapiro-Wilk test was used to determine whether the data conformed to a parametric (Gaussian) or nonparametric (non-Gaussian) distribution. Baseline characteristics and variables were analyzed using the chi-squared test/Fisher’s exact test and independent *t*-test/Mann-Whitney *U* test. The significance of differences in variables between the two groups over time was determined using RMANOVA. Data are presented as mean ± standard-deviation values, and the criterion for statistical significance was *p* < 0.05.

## Figures and Tables

**Figure 1 toxins-13-00077-f001:**
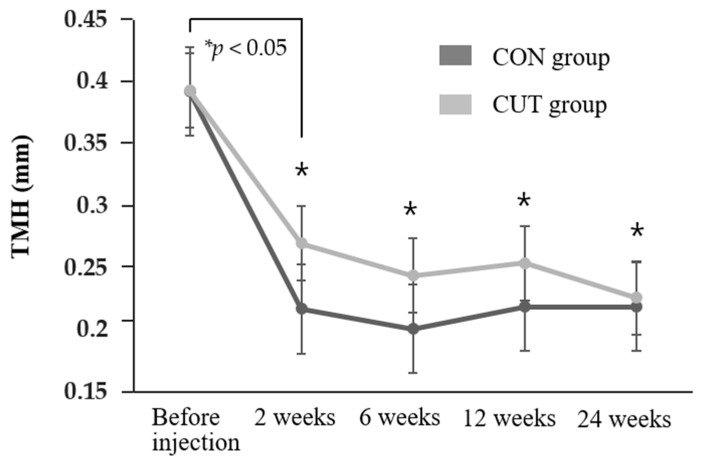
Change in TMH during the 24 weeks of follow-up in the transconjunctival (CON) and transcutaneous (CUT) groups. The tear meniscus height (TMH) decreased significantly relative to before injections (*p* < 0.05) at 2, 6, 12, and 24 weeks. * Repeated-measures analysis of variance (RMANOVA).

**Figure 2 toxins-13-00077-f002:**
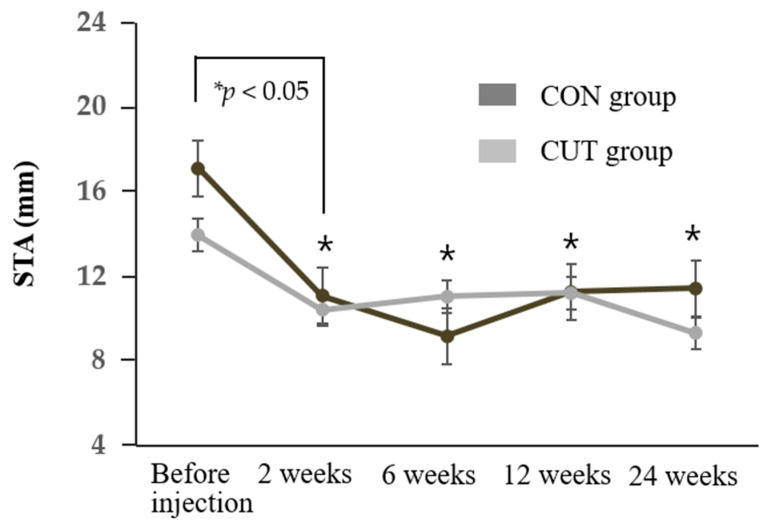
Change in Schirmer’s I test with topical anesthesia (STA) value during the 24 weeks of follow-up in the CON and CUT groups. The STA value decreased significantly relative to before injections (*p* < 0.05) at 2, 6, 12, and 24 weeks. * RMANOVA.

**Figure 3 toxins-13-00077-f003:**
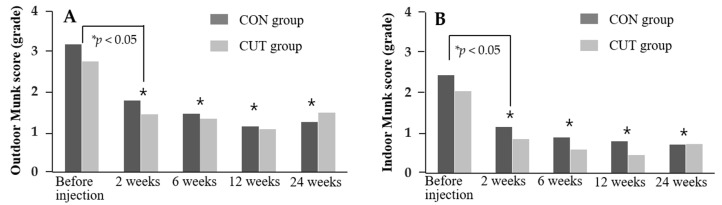
Changes in outdoor and indoor Munk scores during the 24 weeks of follow-up in the CON and CUT groups. Outdoor (**A**) and indoor (**B**) Munk scores decreased significantly relative to before injections (*p* < 0.05) at 2, 6, 12, and 24 weeks. * RMANOVA.

**Figure 4 toxins-13-00077-f004:**
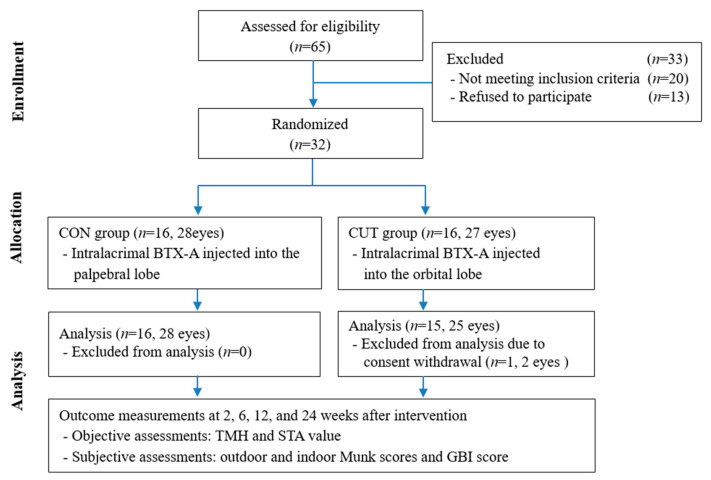
Study flow diagram. GBI—Glasgow Benefit Inventory.

**Figure 5 toxins-13-00077-f005:**
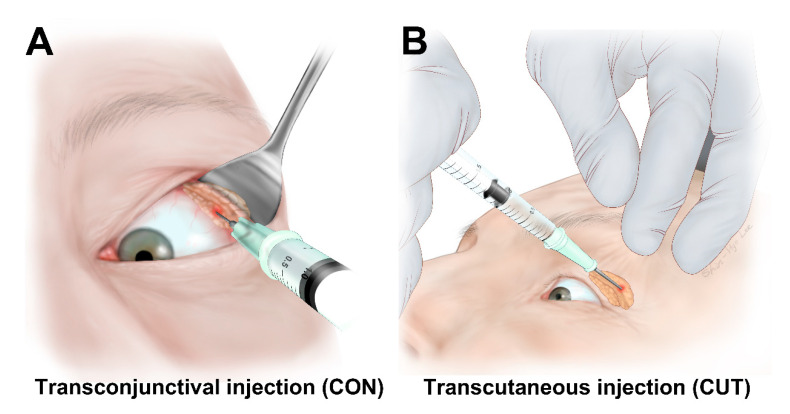
Botulinum toxin A injections into the lacrimal gland. The patient was asked to look inferonasally away from the affected lacrimal gland. (**A**) Transconjunctival injection into the palpebral lobe of the lacrimal gland. (**B**) Transcutaneous injection into the orbital lobe. After entering the skin, the needle was directed superotemporally within the lacrimal gland fossa.

**Table 1 toxins-13-00077-t001:** Demographics and baseline characteristics of the study patients who received botulinum toxin A (BTX-A) via transconjunctival (CON) injections into the palpebral lobe of the lacrimal gland or transcutaneous (CUT) injections into the orbital lobe. TMH—tear meniscus height; STA—Schirmer’s I test with topical anesthesia.

Variable	CON Group (*n* = 16)	CUT Group (*n* = 15)	*p*
Number of eyes	28	25	
Age, years	72.1 ± 9.1	70.2 ± 6.1	0.351 *^a^*
Sex, male:female	6:10	11:4	0.103 *^b^*
Obstruction type, anatomical:functional	8:20	2:23	0.093 *^b^*
Laterality, unilateral:bilateral	4:12	5:10	0.697 *^b^*
TMH, mm	0.39 ± 0.21	0.39 ± 0.15	0.579 *^c^*
STA value, mm	17.11 ± 7.89	13.96 ± 7.73	0.101 *^c^*
Outdoor Munk score	3.17 ± 0.81	2.76 ± 0.92	0.107 *^c^*
Indoor Munk score	2.42 ± 1.29	2.04 ± 1.06	0.226 *^c^*

Data are *n* or mean ± standard-deviation values; *^a^* Independent *t*-test, *^b^* Fisher’s exact test, *^c^* Mann-Whitney *U* test.

## Data Availability

The data presented in this study are available in Appendix A.

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
