# Peer review of "Transconjunctival versus Transcutaneous Injection of Botulinum Toxin into the Lacrimal Gland to Reduce Lacrimal Production: A Randomized Controlled Trial"

_toxins, 2021, doi:10.3390/toxins13020077_

Round 1

Reviewer 1 Report

Review of

Determining the efficacy and optimal delivery route of botulinum toxin into the lacrimal gland of patients with epiphora due to lacrimal outflow obstruction: a randomized controlled trial

Page 1 line 2

The title/heading of the article is too long. The length makes it also less interesting. Something shorter and more intense, for example:

Transconcunctival vs transcutaneus injection of botulinum toxin to reduce lacrimal production: a randomized controlled trial.

Page 1, line 21

Keywords. Do not repeat words included in the title.

Page 1, line 43

Try to make the text more compact, no need to explain everything. Leave out the text highlighted in red. …”especially for surgery candidates in a poor systemic condition such as due to being elderly and having comorbidities such as cerebrovascular diseases and uncontrolled diabetes, and taking an anticoagulant, as well as in patients unwilling to receive surgery [8,9].”

Page2, line 53

Results section now includes information that needs to be in a separate Materials and Methods section preceding (I know the guidelines for Toxins, does not serve the reader) Results section. Include information from Fig 1 and Table 1 in Materials and Methods.

Page 2, line 55.

Abbreviations CON, CUT, TMH, and STA need to be written completely when mentioned the first time and the abbreviation in parenthesis. (they are now explained in Materials and Methods, since the whole section was moved to be after Discussion.)

Page 4, line 71.

Table 1 is unnecessary and should be left out since all the information is in the following Figures and text, so it is duplicated information. If for any reason left in, the heading should be readable alone (without the text), so “after the intervention” should be replaced with the actual intervention, what it was.

Page 4 & 5,line 76,77

Results section is about pure facts, no investigators viewpoints or opinions, so I would leave out words like “tended” or “however”.

Page 6, line 116

First paragraph of Discussion repeats results and should be left out; it is not discussing the results.

Page 7, lines 154-157

In our protocol…” need to be removed to Methods section.

Page 7, line 157

Fig. 1B” Should be Fig 5B.

Page 7, lines 159-161

Avoid repeating the already mentioned results. “Only one case experienced…”

Page 7, line 165

Nava-Castaneda et al [16]. reporting…” Full stop at a wrong spot.

Page 7, line 179

“…the lacrimal gland located than…” Seems to be missing a verb: is, or use other form of expression.

Page 8, line 192

“…Miller21” should be …Miller [19]

Page 8, lines 203-204

Thus, a sufficiently…” Leave out: repeats the exact said in preceding sentence and dulls the culmination end point of the article. Maybe add “preferably” and use “number of patients” to the sentence before: …”maximum duration of the effect in future studies with preferably larger number of patients.”

Page 8 line 211

Even though the Toxins guideline for authors puts the Material and Methods section behind the Discussion, it does not serve the reader. I hope they let you put it back to before Results. If not, you have to change the abbreviations etc in the Results section.

The whole Materials and Methods section to be before Results and include those parts I mentioned before in it from the Results section.

Page 9, line 238

No need to say “…and 6 units in both eye”, “3 units per eye” is enough.

Page 9, line 240

Compress the figure text. Leave out the text highlighted in red, add the text highlighted in yellow.

Figure 5. Botulinum toxin A (BTX-A) injections into the lacrimal gland in the transconjunctival (CON) and transcutaneous (CUT) groups. The patient was asked to look inferonasally away from the affected lacrimal gland. (A) In the CON group, BTX-A was injected Transconjunctivally injection into the palpebral lobe of the lacrimal gland while the lateral upper eyelid was distracted away from the globe. (B) In the CUT group, BTX-A was injected Transcutaneously injection into the orbital lobe. After entering the skin, the needle was directed superotemporally within the lacrimal gland fossa

Page 10, line 269

Complete blood count and chemistry” Meaning/Including what? Why? What was thought to be achieved by doing this?

Page 10, line 287

References are not all according to the guidelines and need to be unified.

Table S1

Bracket in wrong position “6 (week)s” although I don’t see the use of the brackets necessary at all.

Author Response

We are deeply grateful for your sincere and valuable comments, which have resulted in significant improvements to our manuscript. We have revised our manuscript as suggested and have answered all questions to the best of our abilities. We hope our responses satisfactorily address your concerns.

Reviewer 2 Report

Bibliography is incomplete specially in recent years (2017,2018 for exemple). 

No precision about lacrimal obstruction. No dacryoscanner, no lacrimal washout.

What is the difference between functionnal obstruction and patent lacrimal ducts?

Why Schirmer test under anesthesia. It mesures only the basal secretion.

No precision about CUT injection considering the length of the needle.

Some words have to be corrected (errors).

Author Response

(The authors gave the same response as above.)

Round 2

Reviewer 2 Report

About relevant references ther are two articles  in 2017 and 2018 in J Fr Ophtalmol, in english about 27 cases treated with BoNT in lacrimal gland for obstructive epiphora, and 65 cases about epiphora with patent lacrimal ducts that are really missing in this very good article. 

Author Response

: Thank you very much for sharing good papers as references. We have quoted these results in the discussion section, including them in the reference (number 12, 14) as your advice.

(e.g. Girard et al. [12] reported that 90% of patients among 27 cases of obstructive epiphora were very satisfied, with few side effects.)

Thank you again for your sincere and valuable comments. We hope our responses satisfactorily address your concerns.